# GENERATING FURRY CARS: DISENTANGLING OBJECT SHAPE & APPEARANCE ACROSS MULTIPLE DOMAINS

**Utkarsh Ojha**     **Krishna Kumar Singh**     **Yong Jae Lee**

University of California, Davis

`utkarshojha.github.io/inter-domain-gan/`

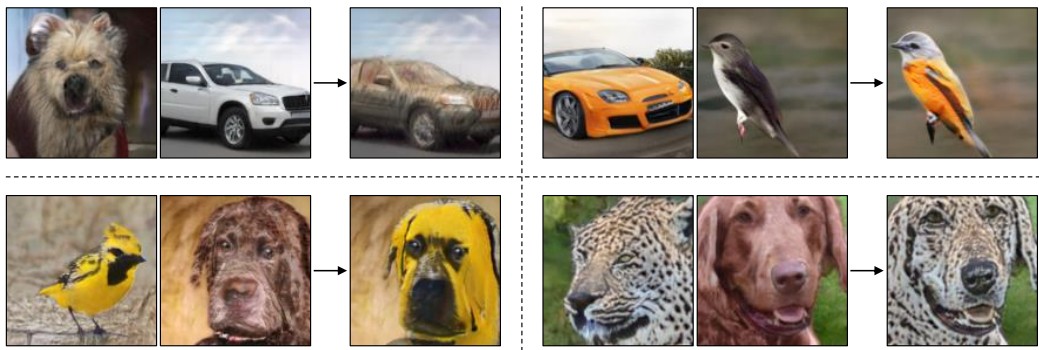

Figure 1: Each block above follows `[Appearance, Shape → Output]`. We propose a generative model that disentangles and combines shape and appearance factors across multiple domains, to create *hybrid* images which do not exist in any single domain.

## ABSTRACT

We consider the novel task of learning disentangled representations of object shape and appearance *across multiple domains* (e.g., dogs and cars). The goal is to learn a generative model that learns an intermediate distribution, which borrows a subset of properties from each domain, enabling the generation of images that did not exist in any domain exclusively. This challenging problem requires an accurate disentanglement of object shape, appearance, and background from each domain, so that the appearance and shape factors from the two domains can be interchanged. We augment an existing approach that can disentangle factors within a single domain but struggles to do so across domains. Our key technical contribution is to represent object appearance with a differentiable histogram of visual features, and to optimize the generator so that two images with the same latent appearance factor but different latent shape factors produce similar histograms. On multiple multi-domain datasets, we demonstrate our method leads to accurate and consistent appearance and shape transfer across domains.

## 1 INTRODUCTION

Humans possess the incredible ability of being able to combine properties from multiple image distributions to create entirely new visual concepts. For example, Lake et al. (2015) discussed how humans can parse different object parts (e.g., wheels of a car, handle of a lawn mower) and combine them to conceptualize novel object categories (a scooter). Fig. 2 illustrates another example from a different angle; it is easy for us humans to imagine how the brown car would look if its appearance were borrowed from the blue and red bird. To model a similar ability in machines, a precise disentanglement of shape and appearance features, and the ability to combine them across different domains are needed. In this work, we seek to develop a framework to do just that, where we define domains to correspond to "basic-level categories" (Rosch, 1978).

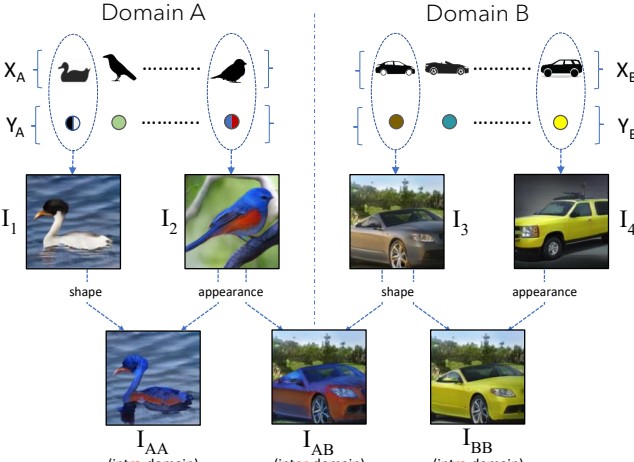

Figure 2: Each domain can be represented with e.g., a set of object shapes ($X_{A/B}$) and appearances ($Y_{A/B}$). The ability to generate images of the form $I_{AA/BB}$ requires the system to learn *intra*-domain disentanglement (Singh et al., 2019) of latent factors, whereas the ability to generate images of the form $I_{AB}$ (appearance/shape from domain A/B, respectively) requires *inter*-domain disentanglement of factors, which is the goal of this work.

Disentangling the factors of variation in visual data has received significant attention (Chen et al., 2016; Higgins et al., 2017; Denton & Birodkar, 2017; Singh et al., 2019), in particular with advances in generative models (Goodfellow et al., 2014; Radford et al., 2016; Zhang et al., 2018; Karras et al., 2019; Brock et al., 2019). The premise behind learning disentangled representations is that an image can be thought of as a function of, say two independent latent factors, such that each controls only one human interpretable property (e.g., shape vs. appearance). The existence of such representations enables combining latent factors from two different source images to create a new one, which has properties of both. Prior generative modeling work (Hu et al., 2018; Singh et al., 2019; Li et al., 2020) explore a part of this idea, where the space of latent factors being combined is limited to one domain (e.g., combining a sparrow's appearance with a duck's shape within the domain of birds; $I_{AA}$ in Fig. 2), a scenario which we refer to as *intra-domain* disentanglement of latent factors. This work, focusing on shape and appearance as factors, generalizes this idea to *inter-domain* disentanglement: combining latent factors from different domains (e.g., appearance from birds, shape from cars) to create a new breed of images which does not exist in either domain ($I_{AB}$ in Fig. 2).

The key challenge to this problem is that there is no ground-truth distribution for the hybrid visual concept that spans the two domains. Due to this, directly applying a single domain disentangled image generation approach to the multi-domain setting does not work, as the hybrid concept would be considered out of distribution (we provide more analysis in Sec. 3). Despite the lack of ground-truth, as humans, we would deem certain combinations of factors to be better than others. For example, if two domains share object parts (e.g., dog and leopard), we would prefer a transfer of appearance in which local part appearances are preserved. For the ones that don't share object parts (e.g., bird and car), we may prefer a transfer of appearance in which the overall color/texture frequency is preserved (e.g. Fig. 2, $I_2$ and $I_{AB}$), which has been found to be useful in object categorization at the coarse level in a neuroimaging study (Rice et al., 2014). Our work formulates this idea as a training process, where any two images having the same latent appearance are constrained to have similar frequency of those low-level features. These features in turn are *learned* (as opposed to being hand-crafted), using contrastive learning (Hadsell et al., 2006; Chen et al., 2020), to better capture the low-level statistics of the dataset. The net effect is an accurate transfer of appearance, where important details remain consistent across domains in spite of large shape changes. Importantly, we achieve this by only requiring bounding box annotations to help disentangle object from background, without any other labels, including which domain an image comes from.

To our knowledge, our work is the first to attempt combining factors from different data distributions to generate abstract visual concepts (e.g., car with dog's texture). We perform experiments on a variety of multi-modal datasets, and demonstrate our method's effectiveness qualitatively, quantitatively, and through user studies. We believe our work can open up new avenues for art/design; e.g., a customer could visualize how sofas would look with an animal print or a fashion/car designer could create a new space of designs using the appearance from arbitrary objects. Finally, we believe that the task introduced in this work offers better scrutiny of the quality of disentanglement learned by a method: if it succeeds in doing so within a domain but not in the presence of multiple ones, that in essence indicates some form of entanglement of factors with the domain's properties.

## 2 RELATED WORK

Learning disentangled representations for image generation has been studied in both the supervised (relying on e.g., keypoints and object masks) (Peng et al., 2017; Balakrishnan et al., 2018; Ma et al., 2018) and unsupervised settings (Li et al., 2018; Shu et al., 2018). Recent work disentangle object shape, appearance, pose, and background with only bounding box annotations (Singh et al., 2019; Li et al., 2020). All prior work, however, focus on disentangling and combining factors within a *single* domain (e.g., birds), and cannot be directly extended to the multi-domain setting since hybrid images would be considered out of distribution (i.e., fake). We present a framework that addresses this limitation, and which works equally well in both single and multi-domain settings.

Another potential angle to tackle the task at hand is through style-content disentanglement (Gatys et al., 2015; Johnson et al., 2016; Ulyanov et al., 2016). However, an object's appearance and shape in complex datasets do not necessarily align with those of style and content (e.g., color of background dominating the style rather than object's appearance). Unsupervised image-to-image translation works (Zhu et al., 2017; Kim et al., 2017; Huang et al., 2018; Gonzalez-Garcia et al., 2018; Choi et al., 2020) translate an image from domain $A$ to domain $B$, such that the resulting image preserves the property common to domains $A$ and $B$ (e.g., structure), and property exclusive to $B$ (e.g., appearance/style). However, if the domains don't have anything in common (e.g., cars $\leftrightarrow$ dogs: different structure and appearance), the translated images typically become degenerate, and no longer preserve properties from different domains. In contrast, our method can combine latent factors across arbitrary domains that have no part-level correspondences. Moreover, when part-level correspondences do exist (e.g., dogs $\leftrightarrow$ tiger), it combines appearance and shape in a way which preserves them. Lee et al. (2018) extended the multimodal image-to-image translation setting by conditioning the translation process on both a content image as well as a query attribute image, so that the resulting output preserves the content and attributes of the respective images. However, this application was explored in settings where both the content and attribute image share similar content/structure (e.g., natural and sketch images of face domain as content/attribute images respectively), which is different from our setting in which the factors to be combined come from entirely different domains (e.g., cars vs birds).

## 3 APPROACH

Given a single dataset consisting of two image domains $\mathcal{A}$ and $\mathcal{B}$ (e.g., dogs and cars), our goal is to learn a generative model of this distribution (with only bounding box annotations and without domain/category/segmentation labels), so that one latent factor (shape) from $\mathcal{A}$ and another factor (appearance) from $\mathcal{B}$ (or vice-versa) can be combined to generate a new out-of-distribution image preserving the respective latent properties from the two domains. Since there is no ground truth for the desired hybrid out-of-distribution images, our key idea is to preserve the frequency of low-level appearance features when transferred from one domain's shape to another domain's shape. To this end, we develop a learnable, differentiable histogram-based representation of object appearance, and optimize the generator so that any two images that are assigned the same latent appearance factor produce similar feature histograms. This leads to the model learning better disentanglement of object shape and appearance, allowing it to create hybrid images that span multiple domains.

We first formalize the desired properties of our model, and then discuss a single domain disentangled image generation base model (Singh et al., 2019) that we build upon. Finally, we explain how our proposed framework can augment the base model to achieve the complete set of desired properties.

### 3.1 PROBLEM FORMULATION

Combining factors from multiple domains requires learning a disentangled latent representation for *appearance, shape, and background* of an object. This enables each of the latent factor's behavior to remain consistent, irrespective of other factors. For example, we want the latent appearance vector that represents red and blue object color to produce the same appearance (color/texture distribution) regardless of whether it is combined with a bird's shape or a car's shape ($I_{AA}$ and $I_{AB}$ in Fig. 2). Henceforth, we represent shape, appearance, and background of an object as $x$, $y$, $b$ respectively, and remaining continuous factors (e.g., object pose) using $z$.

As shown in Fig. 2, we can interpret domain $\mathcal{A}$ as having an associated set of shapes - $X_{\mathcal{A}}$ (e.g., possible bird shapes), and an associated set of appearances - $Y_{\mathcal{A}}$ (e.g., possible bird appear-

ances). With an analogous interpretation for domain $\mathcal{B}$ (e.g., cars), we formalize the following problems: **(i) Intra-domain disentanglement:** where a method can generate images of the form $I = G(x, y, z, b)$, where $[x, y] \in (X_A \times Y_A)$ ($\times$ denotes Cartesian product). In other words, the task is to combine all possible shapes with possible appearances *within* a domain (e.g., $I_{AA}/I_{BB}$ in Fig. 2). **(ii) Inter-domain disentanglement:** where everything remains the same as intra-domain disentanglement, except that $[x, y] \in (X_{AB} \times Y_{AB})$, where $X_{AB} = (X_A \cup X_B)$ and $Y_{AB} = (Y_A \cup Y_B)$. This is a more general version of the task, where we wish to combine all possible shapes with all possible appearances *across* multiple domains (e.g., $I_{AB}$ in Fig. 2).

## 3.2 BASE MODEL

We start with an intra-domain disentangled image generation approach, FineGAN (Singh et al., 2019), as our base model. Fig. 3 (left) provides a high level overview of its architecture, which consists of three image-generation stages: background, shape, and appearance. As a whole, FineGAN takes in four latent variables as input: a continuous noise vector $z \sim \mathcal{N}(0, 1)$, and one-hot vectors $x \sim \text{Cat}(K = N_x, p = 1/N_x)$, $y \sim \text{Cat}(K = N_y, p = 1/N_y)$ and $b \sim \text{Cat}(K = N_b, p = 1/N_b)$, where $N_x$, $N_y$, and $N_b$ are hyperparameters that represent the number of distinct shapes, appearances, and background to be discovered, respectively. Using these, it generates an image in a stage-wise manner: (i) background stage generates a background image ($I_b$); (ii) shape stage creates a shape image ($I_x$) by drawing the silhouette of the object over the background using a shape mask ($m_x$); (iii) the appearance stage generates the color/texture details, and stitches it to $I_x$ using an appearance mask ($m_y$) to create the final image ($I$).

To learn to disentangle object from background, FineGAN relies on bounding box supervision. To learn the desired disentanglement between shape and appearance without any supervision, FineGAN uses information theory to associate each factor to a latent code, and constrains the relationship between the codes to be hierarchical. Specifically, it makes the assumption that $N_x < N_y$; i.e., variety in object shape is less than variety in object appearance, which is generally true for real-world objects (e.g., different duck species have different colors/textures but share the same overall shape). During training, this hierarchy constraint is imposed so that a set of latent appearance codes $y$ will always be paired with a particular latent shape code $x$. This ensures that, during training, the shape code representing e.g., a duck's shape does not get paired with the appearance code for e.g., a sparrow's texture to create an unnatural combination (which the discriminator could use to easily classify the generated image as fake, and in turn prevent the model from learning the desired disentanglement). Two types of loss functions are used. An adversarial loss $L_{adv}$ (Goodfellow et al., 2014) to enforce realism on the background $I_b$ (on a patch-level) and final generated image $I$ (on an image-level), and a mutual information loss $L_{info}$ (Chen et al., 2016) between (i) $x$ and *masked* object shape ($m_x * I_x$), (ii) $y$ and *masked* object appearance ($m_y * I$), to help the latent codes gain control over the respective factors:

$$\mathcal{L}_{adv} = \min_G \max_D \ \mathbb{E}_r[\log(D(r))] + \mathbb{E}_c[\log(1 - D(G(c)))]$$
$$\mathcal{L}_{info} = \max_{D,G} \ \mathbb{E}_c[\log D(c | m * G(c))]$$

where $c$ represents the input latent codes, $r$ is the distribution of real patches/images, $G/D$ are the generator and discriminator, $G(c)$ is the generated image, and $m$ is the generated mask. For the background stage, only $\mathcal{L}_{adv}$ is used on a patch-level, where $r$ is the distribution of real patches and $c = \{z, b\}$. For the shape stage, only $\mathcal{L}_{info}$ is used, with $c = x$ and $m = m_x$. Finally, for the appearance stage, both $\mathcal{L}_{adv}$ and $\mathcal{L}_{info}$ are used, where $r$ is the distribution of real images and $c = \{z, b, x, y\}$ for $\mathcal{L}_{adv}$, and $c = y$ and $m = m_y$ for $\mathcal{L}_{info}$. We denote all the losses used by the base model (FineGAN in this case) as $L_{base}$. Fig. 3 (bottom) summarizes the properties learned by the base model.

## 3.3 COMBINING FACTORS FROM MULTIPLE DOMAINS

Although the base model can combine shape and appearance within a single domain (Fig. 2 bottom left), it has trouble doing so across different ones, as illustrated in Fig. 4 (b) top, where the same latent appearance code behaves differently when combined with a car's shape ($I$) versus a bird's shape ($I_{pos_2}$). The main reason is because hybrid images, which combine e.g., a bird's colors with a car's shape, are non-existent in the real data (i.e., out of distribution). Thus, the model is penalized from

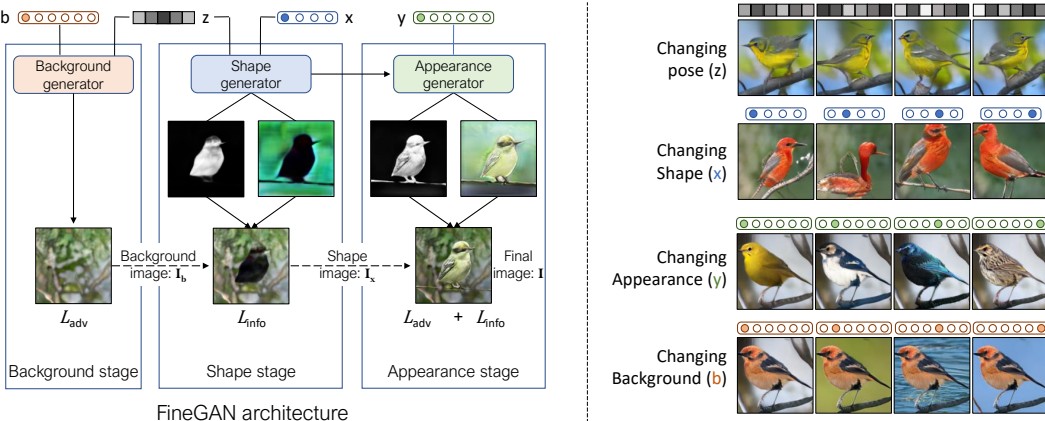

Figure 3: **Left:** A simplified model architecture of the base model FineGAN (Singh et al., 2019), where images are generated in a stagewise manner, by generating the background, shape, and appearance of the object, in that respective order. Each of the latent codes controls certain properties about the image (e.g. $x$ code controlling the shape). **Right:** The *intra*-domain disentanglement capability of the base model. Each block demonstrates the effect of changing one latent code (shown above each image), which controls a single factor in the generated image.

learning such a combination via the adversarial loss. For creating hybrid objects within a domain (e.g., combining red sparrow color and seagull shape to create a non-existent red seagull), this is less of a problem, since the base model implicitly learns part semantics as it has to consistently colorize the correct part of an object regardless of its pose (i.e., it learns a pose-equivariant representation of appearance; see Fig. 3 top-left). And since intra-domain objects share part semantics, during test time (in which the adversarial loss no longer plays a role), transferring e.g., a sparrow's appearance to a seagull's shape can still be achieved with some degree of success. However, when the two domains do not share any part semantics (e.g., birds and cars), the model would not know where e.g., the color of a bird's head should go on a car. In other words, when trained in a multi-domain setting, the base model is unable to fully disentangle shape from appearance; i.e., there is still some form of entanglement of shape and appearance tied to each domain.

So, how can we learn better disentanglement between appearance and shape in the multi-domain setting, without any ground truth for hybrid images? Our idea is to preserve the *frequency* of low-level visual concepts (color/texture) as a heuristic when transferring the appearance from one domain to another, and not transfer any shape information in the process (e.g. transfer a car's color and texture distribution to a dog, while preventing transfer of its wheels). Specifically, any two generated images with the same latent appearance code (but different shape codes) should have similar frequency in low-level appearance. To use this idea as part of an optimization process, we need to: (i) define the concepts whose frequency needs to be computed and make its computation differentiable, and (ii) induce the generator to generate hybrid images that preserve the low-level appearance frequency.

**Learning a differentiable histogram of low-level features.** At a high-level we'd like texture and color frequency to be our low-level feature concepts. However, since it is difficult to know a priori which specific textures/colors would be useful (e.g., a feature representing leopard texture would be useful for shape/appearance disentanglement in animals ↔ cars, but might not be for birds ↔ cars), we propose to learn them in a data driven way. Specifically, we represent the low-level concepts using a set of learnable convolutional filters (Fig. 4(a)), each of which convolves over a generated image to give regions of high correlation. To focus on the object region, we mask out the background using the base model's generated foreground mask, and compute the channel-wise sum of the resulting response maps to get a $k$-dimensional histogram representation $h$. This representation approximates the frequency of visual concepts represented by the set of filters (e.g., dog's fur, tiger's texture).

How do we ensure that the histogram $h$ being approximated is of some relevant object feature? The information captured by the filter bank should be such that (i) the same object in different pose/viewpoints has a similar appearance representation, (ii) which in turn should be dissimilar than

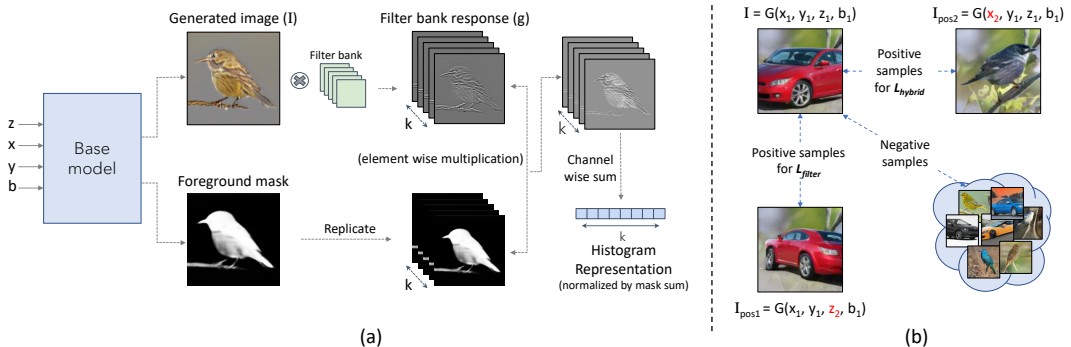

Figure 4: **(a)** The process of computing the frequency-based representation. All steps are differentiable, making the representation learnable as part of the optimization process. **(b)** We construct positive samples differently for $L_{filter}$ and $L_{hybrid}$, whereas negatives remain the same for both.

the representation of any other arbitrary object. We hence train the filter bank using contrastive learning (Bachman et al., 2019; Chen et al., 2020). Specifically, we first construct a batch of $N$ generated images, where each image is of the form $I = G(x_1, y_1, z_1, b_1)$. We then construct another batch, where each image is of the form $I_{pos_1} = G(x_1, y_1, z_2, b_1)$, by only altering the noise vector $z$, which changes only the object's pose (due to our base model; Fig. 3 top-left). We set each $I$ and corresponding $I_{pos_1}$ pair as positive samples, and all other image pairs as negative samples (Fig. 4(b)). Finally, we use $NT\text{-}Xent$ (Normalized temperature-scaled cross entropy loss (Chen et al., 2020)) for each image $I$ in the batch to learn the filter bank:

$$\ell_i = -\log \frac{\exp\left(\text{sim}\left(h_i, h_j\right)/\tau\right)}{\sum_{k=1}^{2N} \mathbf{1}_{[k \neq i]} \exp\left(\text{sim}\left(h_i, h_k\right)/\tau\right)} \tag{1}$$

where $h_i$ represents the feature histogram of $i^{th}$ image, $j$ and $k$ indexes the positive and negative samples for $i$, respectively, $\text{sim}$ denotes cosine similarity, and $\tau$ is the temperature hyperparameter (set to 0.5). We refer to the whole loss as $L_{filter} = \sum_{i=1}^{N} \ell_i$, and optimize it by only updating the weights of the filter bank.

**Conditioning the generator to generate hybrid images.** Now that we have a way to learn meaningful low-level features in a data driven way, we can use them to condition the generator to generate hybrid images. We use an objective similar to $L_{filter}$, where we use the same $I = G(x_1, y_1, z_1, b_1)$. However, we construct the positive pairs differently: $I_{pos_2} = G(x_2, y_1, z_1, b_1)$, in which the latent shape code $x$ is different than that of $I$ (while all other codes are the same); see Fig. 4(b). We refer to this loss as $L_{hybrid} = \sum_{i=1}^{N} \ell_i$, and optimize only the generator $G$'s parameters. Intuitively, $L_{hybrid}$ pushes the generator to match the frequency of low-level appearance features (e.g., car's color) between the generated images of two arbitrary shaped objects (e.g., car and bird).

The overall loss function of our framework is:

$$\mathcal{L} = L_{base} + L_{hybrid} + L_{filter} \tag{2}$$

where $L_{base}$ induces the properties needed for intra-domain disentanglement, and the combination of $L_{filter}$ and $L_{hybrid}$ helps extend the abilities to learn inter-domain disentanglement of factors. Note that we do not employ the adversarial loss (which is part of $L_{base}$) on any hybrid images that do not follow our base model's hierarchical code constraints (as described in Sec. 3.2) since those images would be outside the real data distribution.

## 4 RESULTS

We now compare the proposed framework against several baselines, and study how they fare in terms of disentangling and combining shape/appearance from different domains.

We use three single domain datasets: CUB (Wah et al., 2011) collection of 200 birds categories. (ii) Stanford Dogs (Khosla et al., 2011) collection of 120 dog categories. (iii) Stanford Cars (Krause et al., 2013) collection of 196 car categories. Using these, we construct three multi-domain settings:

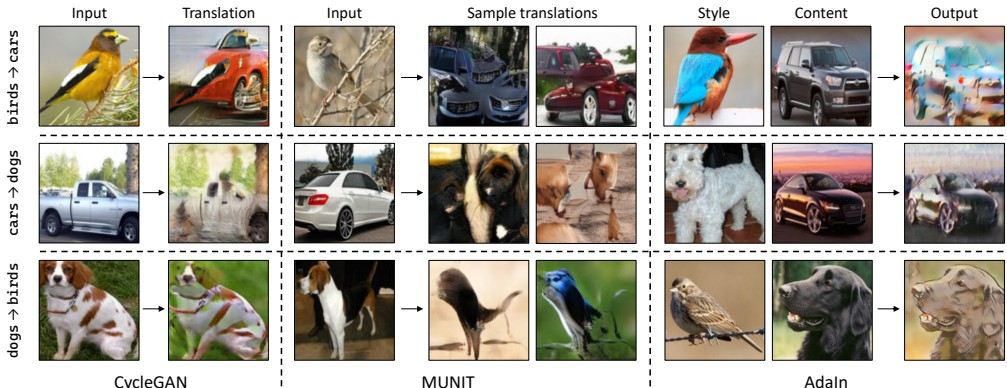

Figure 5: Results using CycleGAN, MUNIT, and AdaIn. When the two domains don't have structural correspondence, the image-to-image translation methods fail to disentangle shape from appearance. AdaIn suffers, as images don't always have a homogeneous style to be extracted from, resulting in misalignment in the definition of object appearance and style.

(i) birds $\leftrightarrow$ cars, (ii) cars $\leftrightarrow$ dogs, (iii) dogs $\leftrightarrow$ birds, consisting of a total of 396, 316, and 320 object categories, respectively. We also use a multi-domain dataset of animal faces (Liu et al., 2019), comprising 149 carnivorous categories, where each category can be interpreted as a different domain. This forms our fourth multi-domain setting, (iv) animals $\leftrightarrow$ animals.

## 4.1 CAN IMAGE TRANSLATION OR STYLE TRANSFER METHODS SOLVE THIS TASK?

As mentioned in Sec 2, there are existing works which could potentially offer solutions for the task at hand, even with their core objective being different. CycleGAN (Zhu et al., 2017) and MUNIT (Huang et al., 2018) translate images from one domain to another, in a way where the translated images preserve some properties from both the source and target. If the two domains are, say cars$\leftrightarrow$dogs, can these methods translate a car into an image that combines the car's shape with a dog-like appearance? Fig. 5 shows some translation results: in general, we observe that the translations become degenerate, and don't preserve any interpretable property from the input, i.e., the methods are no longer able to disentangle structure from domain-specific appearance. We also explore AdaIn (Huang & Belongie, 2017), a method capable of transferring *style* between any arbitrary pair of images; see Fig. 5. We notice that the definition of *style* transferred and what we perceive as appearance don't always align, and even then, the method is unable to solely focus on the foreground object when extracting and transferring the style (e.g., bird's blue color bleeds into the background of the car in the first row). Overall, Fig. 5 shows that these methods are not suited for settings involving arbitrary domains, which have no similarity in object shape/appearance.

## 4.2 COMPARISON TO MULTI-FACTOR DISENTANGLEMENT BASELINES

We next study baselines which disentangle object shape, appearance, pose, and background, and generate images conditioned on the respective latent vectors: **(i) FineGAN** (Singh et al., 2019): base model described in Sec. 3.2; **(ii) Relaxed FineGAN**: an extension of FineGAN, where we relax the hierarchical constraint between shape and appearance codes (so that at least conceptually, hybrid images can be generated during training). For those hybrids, we only optimize $L_{info}$ whereas for the remaining images (generated with code constraints), we optimize the full $L_{base}$. **(iii) Ours w/o** $L_{filter}$: our approach using a fixed, randomly initialized filter bank to compute the frequency-based representation; **(iv) Ours**: our final approach. Note that we don't need domain labels to train methods (i)-(iv); but once trained, we can visualize what the different learned shape ($x$) and appearance ($y$) latent vectors represent (e.g., $x_1$ and $x_2$ might generate a specific bird and car shape, respectively). Hence, for evaluation purposes, we manually create a split of the latent vectors, $[X_A, Y_A]$ and $[X_B, Y_B]$, so that $\forall x \in X_A$, $x$ represents only one domain (e.g., birds' shapes). This way, we can combine latent factors from different domains to study shape/appearance disentanglement.

We first show qualitative comparisons to FineGAN over different multi-domain datasets in Fig. 6. FineGAN has issues combining factors from different domains, evident through (i) inaccurate ap-

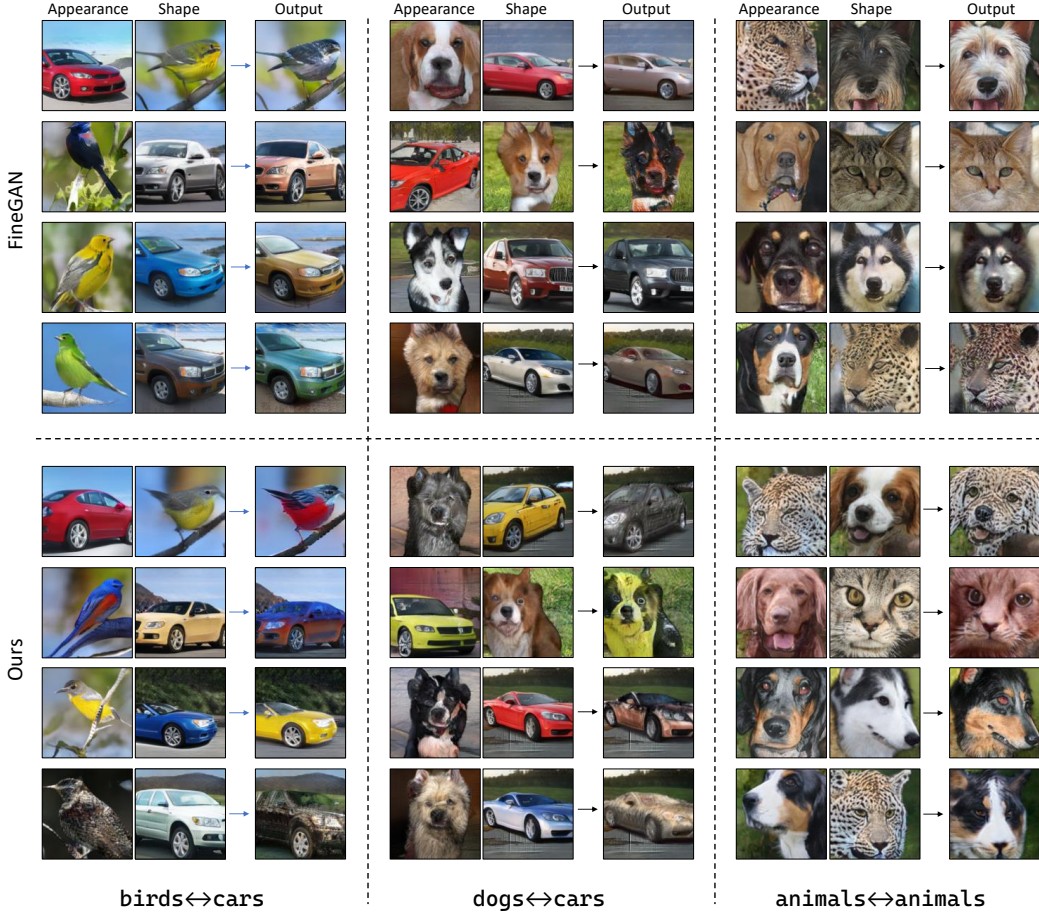

Figure 6: **FineGAN (top) vs. Ours (bottom)**. The only difference between the 'Output' and 'Shape' image is the latent appearance vector, which is borrowed from the 'Appearance' image. Our outputs preserve the characteristic appearance details much better than FineGAN. More results in Sec. A.6.

|  | birds $\leftrightarrow$ cars | | cars $\leftrightarrow$ dogs | | dogs $\leftrightarrow$ birds | | animals $\leftrightarrow$ animals | |
|---|---|---|---|---|---|---|---|---|
|  | color | texture | color | texture | color | texture | color | texture |
| FineGAN (Singh et al., 2019) | 0.4295 | 0.4234 | 0.3856 | 0.4006 | 0.2862 | 0.3442 | 0.1914 | 0.2322 |
| Relaxed FineGAN | 0.3121 | 0.4005 | 0.3451 | 0.3912 | 0.2433 | 0.3287 | 0.1214 | 0.1834 |
| AdaIn (Huang & Belongie, 2017) | 0.5619 | 0.4476 | 0.5113 | 0.3956 | 0.6265 | 0.3512 | 0.2447 | 0.1801 |
| Ours w/o $L_{filter}$ | 0.2842 | 0.3341 | 0.2289 | 0.2754 | 0.2228 | 0.2744 | 0.0721 | 0.0815 |
| Ours | **0.2484** | **0.3057** | **0.2069** | **0.2535** | **0.2009** | **0.2573** | **0.0462** | **0.0695** |

Table 1: $\chi^2$-distance between color/texture histograms of source and target images (lower is better). By enforcing similarity in the frequency of low-level features (via $L_{filter} + L_{hybrid}$), our model better retains the color/texture distribution across the source and target compared to alternate baselines.

pearance transfer (e.g., row 1: only car window color gets transferred to the bird), and (ii) the inability to transfer details beyond rough color (e.g., row 4: the fur texture of the brown dog isn't transferred to the car; row 1: the characteristic leopard texture isn't transferred to the dog). Our method, on the other hand, results in a more holistic transfer of appearance to arbitrary shapes. Upon transferring, it is able to better preserve the (i) color distribution (row 5/6: both the blue and red components get transferred to the car/bird respectively), (ii) other low-level characteristic features (row 8: brown fur gets transferred to the car's body; row 5: leopard's texture is transferred to the dog's face). Focusing on animals $\leftrightarrow$ animals results for FineGAN, we see that the *Shape* images have a tendency to resist appearance change (e.g., the leopard's texture is still preserved in the output image), which indicates there is some entanglement between shape and appearance. This is mainly because unlike our approach, FineGAN does not have an explicit constraint to enforce the same latent appearance code to produce the same low-level visual features regardless of shape. We study this property in more detail in Sec. A.3.

|  | Shape disentanglement | | AMT results | |
| --- | --- | --- | --- | --- |
|  | FineGAN | Ours | Ours vs AdaIn | Ours vs FineGAN |
| birds ↔ cars | $0.831 \pm 0.04$ | $0.861 \pm 0.03$ | $82.56 \pm 3.28$ | $75.04 \pm 4.67$ |
| cars ↔ dogs | $0.835 \pm 0.03$ | $0.841 \pm 0.03$ | $85.45 \pm 5.11$ | $83.02 \pm 4.55$ |
| dogs ↔ birds | $0.674 \pm 0.10$ | $0.748 \pm 0.08$ | $75.05 \pm 6.89$ | $63.25 \pm 7.17$ |
| animals ↔ animals | $0.871 \pm 0.06$ | $0.907 \pm 0.02$ | $82.51 \pm 1.85$ | $76.21 \pm 2.12$ |

Table 2: **(Left)** Shape disentanglement results: higher IoU means better shape disentanglement from appearance. **(Right)** AMT experiment results: how often our method is preferred over the baseline.

### 4.2.1 QUANTIFYING SHAPE/APPEARANCE DISENTANGLEMENT

**Appearance transfer:** In this experiment, we evaluate how well appearance is transferred from one domain to another. For each latent appearance vector $y_i$, we generate a source $I_s = G(x_i, y_i, b_i, z_i)$ and target image $I_t = G(x_j, y_i, b_i, z_i)$ ($x_i, x_j$ represent shapes from different domains). We segment the foreground pixels using DeepLabv3 (Chen et al., 2017), pre-trained on COCO (Lin et al., 2014). We then evaluate the similarity between the source and target images' foreground appearance by computing the $\chi^2$-distance between their color and texture histograms (Leung & Malik, 2001) (more details in Sec. A.4.1). Table 1 summarizes the results. As expected based on Figs. 5 and 6, the baselines often have issues accurately transferring the color/texture distribution to an arbitrary shape, resulting in the color/texture histogram for $I_s$ and $I_t$ having low similarity. 'Ours w/o $L_{filter}$' is a decent baseline, which simply uses random filters to minimize $L_{hybrid}$. Our final approach, which learns the filters, achieves the best overall performance in retaining the color and texture information.

**Shape transfer:** We next evaluate how well the object's original shape is retained when appearance from another domain is transferred to it. If shape is accurately disentangled from appearance, then if we generate a stack of images $\{G(x, y_i, z, b)\} \vee y_i \in$ S (arbitrary set of latent appearances), any differences should only be in the foreground appearance (Fig. 3 top right). We therefore compute the standard deviation across the stack, and convert it into a binary mask using an appropriate threshold. If different sets of appearances (S) result in similar masks, that is an indication of shape being controlled only by $x$. So, we randomly split the available latent vectors into two sets, $S_1$ and $S_2$, compute their respective masks, and evaluate the IoU between them (Sec. A.4.2). This is repeated for 10 different random splits of $S_1/S_2$, giving the score for a latent shape. The final score is averaged over all possible latent shapes. Table 2 (left) summarizes the results: we see that both FineGAN and our method achieve high IoU scores, demonstrating that changes introduced by different appearance vectors are mostly confined within the same region, and hence *don't* change the object shape.

**Hybrid images through the eyes of image classifiers:** If we look at the last row in Fig. 6, a regular car becomes a car with brown fur. For an image classification network, does this *furry-car* have more dog-like properties, and less of cars'? When compared to the hybrids created by FineGAN, we find our hybrids do indeed exhibit this behavior; see Sec A.5 in Appendix for details.

**Perceptual study:** Finally, we conduct human studies. Each task shows the results of two methods, one of which is ours and the other is either AdaIn or FineGAN. The result format is same as in Fig. 6, i.e., [Appearance, Shape → Output] (Appearance/Shape images are used as Style/Content input for AdaIn). Amazon Mechanical Turkers are asked to select the method which resulted in a better shape/appearance transfer. We generate 250 tasks for each dataset, and gather responses from 5 unique turkers for each task, resulting in 1250 judgements. Table 2 (Right) summarizes the results. Averaged across all datasets, our method gets chosen over AdaIn and FineGAN in $81.39 \pm 3.85\%$ and $74.38 \pm 7.11\%$ of the cases, demonstrating consistent superiority in generating hybrid images.

## 5 DISCUSSION AND CONCLUSION

The proposed method takes a step towards learning inter-domain disentanglement through the ability to combine shape and appearance across domains. However, there are some limitations: our method builds upon FineGAN, which in turn makes some assumptions about the data (existence of a hierarchy between shape and appearance factors). Consequently, our method is suited for datasets with similar properties. Furthermore, it becomes computationally expensive to train models on all combinations with increasing domains ($\binom{n}{2}$, which grows quadratically). Nonetheless, we believe that we address an important and unexplored problem having practical applications in art and design.

**Acknowledgements.** This work was supported in part by a Sony Focused Research Award and NSF CAREER IIS-1751206.

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

# A APPENDIX

## A.1 TRAINING DETAILS

Since we use FineGAN as our base model, the generator and discriminator architecture, as well as hyperparameters like learning rate, optimizer (Adam (Kingma & Welling, 2014)), $\beta_1/\beta_2$ are borrowed from (Singh et al., 2019). We also use the same data preprocessing step, where we crop the images to $1.5 \times$ their available bounding box, so as to have a decent object to image size ratio. We use a batch size of 24 to maximize the usage of two TITAN Xp GPUs per experiment. Apart from the raw data, FineGAN uses two forms of information to disentangle object pose, shape, appearance and background while modeling an image distribution - (i) bounding box information, so that patches outside it can be used as real background patches to model background on a patch level; (ii) No. of object categories in the dataset. The hierarchy to be discovered needs to be decided beforehand, where the number of distinct appearance ($N_y$) are set to be the number of object categories, and the number of shapes ($N_x$) is set empirically. We set $N_y$ as 400 ($\sim$200 + 196) for birds $\leftrightarrow$ cars, 320 ($\sim$196 + 120) for cars $\leftrightarrow$ dogs and 320 (120 + 200) for dogs $\leftrightarrow$ birds. We approximate the 196 car categories as 200, since it helps enforce the hierarchy constraints more easily. Furthermore, $N_x$ = $N_y$/10 for all these three datasets. Note that this was the same configuration used by FineGAN ($N_x = N_y$/10), which the authors found to be a good default relationship to disentangle shape and appearance in multiple datasets. The original paper also studies the sensitivity of the generated images' quality to the choice of $N_x/N_y$, and found that the results are largely agnostic to these choices (except when those two values become very skewed: e.g. $N_x = 5$). For animals $\leftrightarrow$ animals, we set $N_y$ as 150 ($\sim$149 categories), and $N_x$ as $N_y$/5.

Upon training FineGAN, the four different latent vectors start to correspond to some properties in the generated images: latent appearance vector, having a one-hot representation, starts capturing the appearance (different indices start generating different appearance details) etc. This way, one can combine the latent shape code which generated duck, latent appearance code which generated yellow bird, to create a yellow duck. For birds $\leftrightarrow$ cars, cars $\leftrightarrow$ dogs, and dogs $\leftrightarrow$ birds, we first train the base model until convergence (600 epochs). This model can disentangle disentangle the four factors within a domain as shown in Fig. 3. Starting from this pre-trained model, we now use our loss objectives $L_{filter}$ and $L_{hybrid}$, in addition to $L_{base}$ (FineGAN's overall objective), and further train the model to better perform inter-domain disentanglement of factors.

Specifically, in each iteration, we optimize the model using $L_{base}$, then optimize the generator's weights using $L_{hybrid}$, and then optimize the filters' weights using $L_{filter}$. In our experiments, we found that using $L_{hybrid}$ at lesser frequency (1/4th of the time) compared to $L_{base}/L_{filter}$ works better. The second phase of training is performed for about (50k iterations). The two phase training for these three datasets is important, because we observed that directly using the combination of $L_{base}$, $L_{hybrid}$ and $L_{filter}$ while training from scratch leads to model having some issues with learning intra-domain disentanglement. For the fourth dataset (animals $\leftrightarrow$ animals), however, we directly train from scratch using all three loss components, a setting which actually results in better performance than the two phase one discussed above. This difference could be attributed to the difference in the nature of multi-domain datasets, where birds $\leftrightarrow$ cars has only two domains, the animals $\leftrightarrow$ animals domain effectively has $N_p$ (= 30) domains.

## A.2 ABLATION STUDY

We now study different design choices, and visualize their effects in mixing shape/appearance across domains.

**Receptive field size of filters:** As explained before, we can use more than one layer of filter while computing the frequency-based representation. As shown in Fig. 4 (left), one can have a second layer of filters which take as input the filter bank responses (g), and repeats the same process again. The final frequency based representation will be the concatenation of representation after first and second stage. In our experiment, we always use filters of size 3x3 (unless stated otherwise), stride = 1, padding = 0. The number of filters in the first, second and third layers are 64, 128 and 192 respectively. We experiment with 4 settings - (i) 1 layer of 1x1 filters; (ii) 1 layer of 3x3 filters; (iii) 2 layers of 3x3 filters; and (iv) 3 layers of 3x3 filters. Between each layer is a combination of ReLU non-linearity and maxpool operation (which further helps in increasing the effective receptive

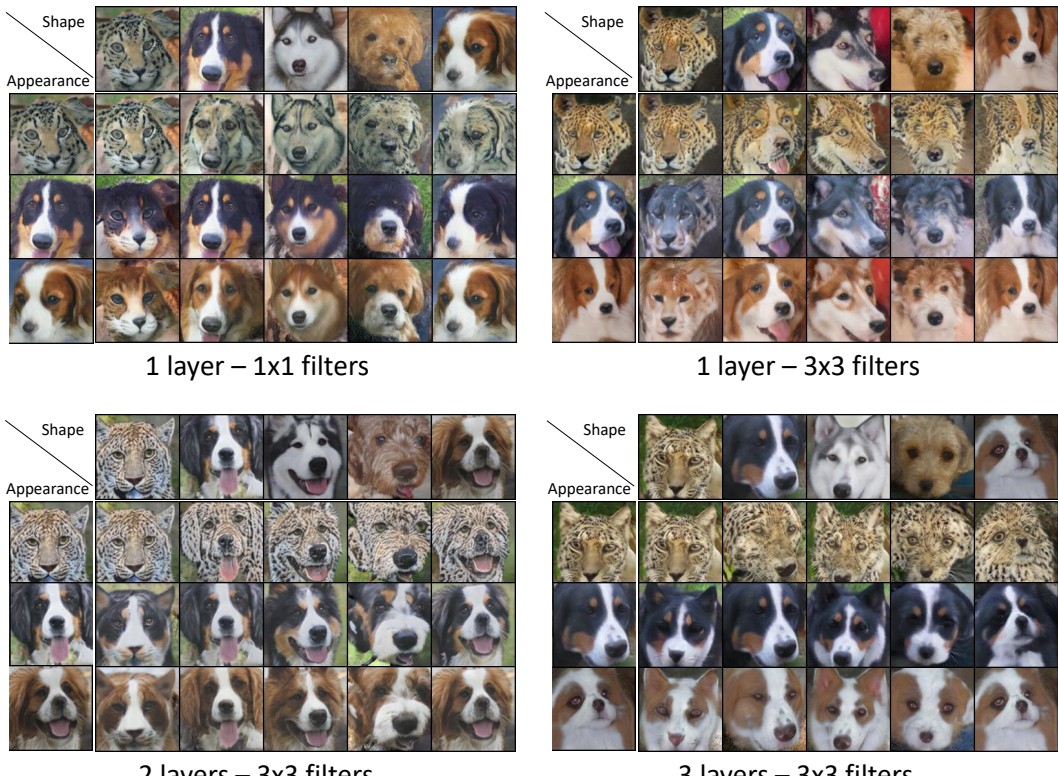

Figure 7: Results of our method with different settings of filters used to approximate frequency-based representation. As the effective receptive field size increases from top-left to bottom right, we notice that the method can better transfer characteristic appearance details.

field). The results are shown in Fig. 7. With setting (i), i.e. 1 layer of 1x1 filters, the only additional constrain introduced by $L_{filter}$ and $L_{hybrid}$ is to ensure consistency in 1x1 regions, i.e. RGB pixel distribution. In other words, this setting could be thought of as enforcing color histogram similarity between source and target images. We observe a similar behavior in the results as well, where it fails to accurately transfer the tiger's texture to other animals' faces, but is able to faithfully transfer the color distribution. Furthermore, the source dog in the second row has two characteristic property; brown patches above the eyes, and a white region in the middle going all the way to the forehead. Transferring this particular appearance doesn't always preserve the properties; e.g. (i) the tiger only has a white patch near the mouth, which doesn't extend towards the forehead, (ii) the brown patches over the eyes are no longer transferred to the tiger's face. In general, we observe that these properties (white patch, texture pattern), which extend beyond color distribution, are better captured with increasing receptive field. Setting (iii) and (iv) result in similar performance, so we choose the setting (iii) (a simpler version) as our final model.

**Parameters optimized using $L_{hybrid}$:** We explain in sec. 3.3 that $L_{hybrid}$ is optimized by adjusting the generator weights. However, the base model that we use (FineGAN) generates the foreground in two stages: the shape stage generating the outline, and the appearance stage filling in the details. In animals $\leftrightarrow$ animals, the shape stage often captures some appearance details as well (e.g. the leopard's texture), and hence only optimizing the appearance stage doesn't let the model transfer those characteristic details, since they are not being generated at the appearance stage (Fig. 8). Optimizing both the shape and appearance stage helps get rid of that issue, where the generation of appearance details gets properly shifted to the appearance stage, resulting in more accurate appearance transfer (Fig. 8 right).

**Effect of $\tau$:** $NT\text{-}Xent$ (Normalized temperature-scaled cross entropy loss), which is used in both $L_{hybrid}$ and $L_{filter}$, uses $\tau$ as the temperature hypermarameter. For all the experiments up until now,

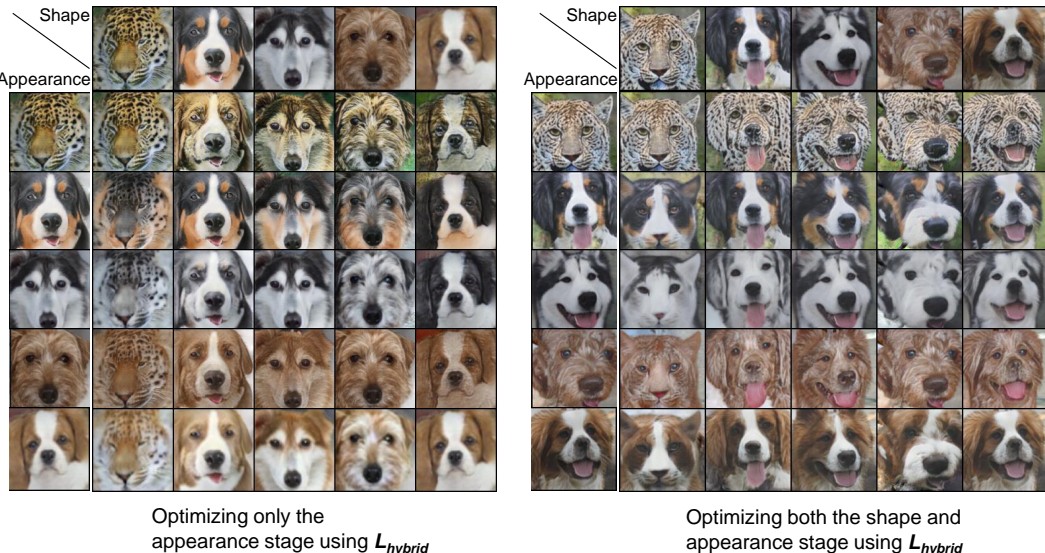

Figure 8: Analysis of the effect of training only the appearance stage vs both shape and appearance stage.

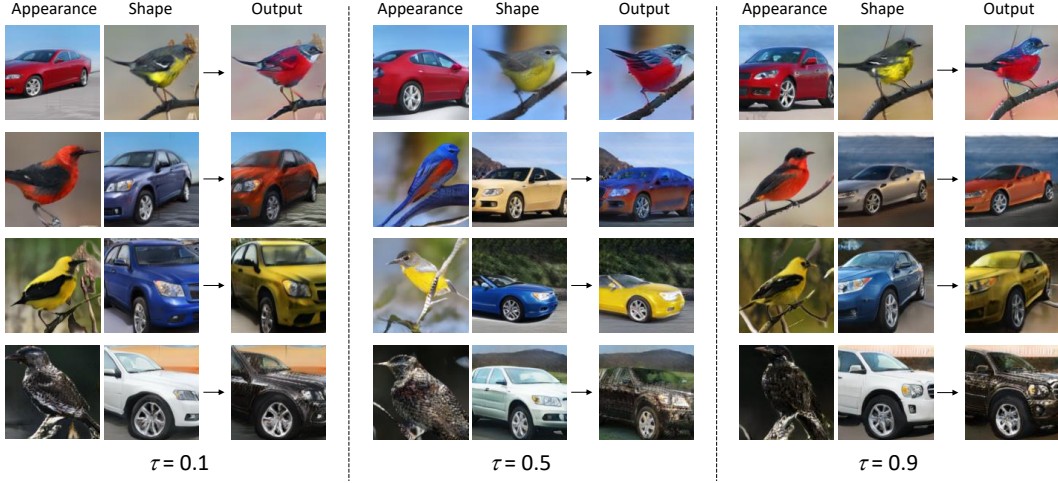

Figure 9: Effect of the temperature parameter ($\tau$), used in $L_{hybrid}$ and $L_{filter}$

we choose $\tau = 0.5$ as the default value. In this section, we study whether our method's performance is sensitive to the value used for $\tau$. Results are presented in Fig. 9, where we consider the birds $\leftrightarrow$ cars setting, and run our algorithm for three values of $\tau$, 0.1, 0.5 and 0.9. We observe that the ability to accurately transfer the appearance from one object to another remains largely consistent across the three cases. This indicates that our method is flexible, where one doesn't need to tune the value of $\tau$ for accurate inter-domain disentanglement.

## A.3 RESISTIVITY ISSUE

In Fig. 10, we see that FineGAN shows a *resistance* towards changing the appearance of an animal; e.g. in row 4, the leopard in the target image only borrows a blackish color tone, and most of the signature leopard appearance remains as it is. In contrast, our method is able to comprehensively change the appearance of an animal; e.g. in the same row, leopard can shed its appearance and borrow everything from the tricolored dog, while also preserving any potential correspondence (the resulting animal retains the brown patch above the eye). We study the resistivity property in more detail: for each latent shape vector $x_i$, we construct a list of images ($L$) by combining it with

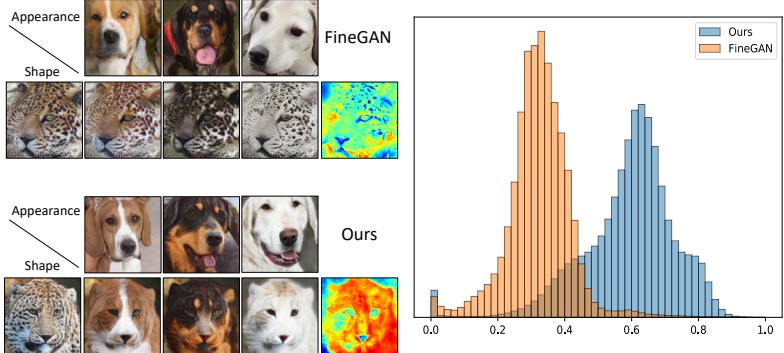

Figure 10: Analysis of the resistance of a model to alter the appearance. Generations using FineGAN don't fully relinquish the appearance, resulting in a low standard deviation for pixel changes across the spatial locations compared to our approach.

all possible latent appearance vectors, keeping the latent noise ($z$ capturing pose) and background vectors the same, i.e. $L = \{G(x_i, y_j, z_i, b_i)\} \vee y_j \in$ **animals** $\leftrightarrow$ **animals**. We then stack the images in $L$, compute standard deviation across channels separately, and then average to generate a heatmap image. We repeat this process for all the latent shape vectors, and plot the histogram of standard deviation values. Fig. 10 illustrates this process, and compares the heatmap images generated using our approach vs FineGAN. This helps us visualize the *resistance* property, as the heatmap and histogram through FineGAN indicates much lower standard deviation of pixel differences compared to our approach, implying more diversity of visual appearance attainable using our method.

## A.4 SHAPE/APPEARANCE DISENTANGLEMENT

### A.4.1 COLOR/TEXTURE HISTOGRAM CONSTRUCTION DETAILS

We first define the process of obtaining the image and texture based cluster centers (codebook), and then detail the process of constructing and comparing the respective histogram representations of two images.

We sample 50,000 pixel (RGB) values randomly from the images belonging to a multi-domain dataset (e.g. **birds** $\leftrightarrow$ **cars**), and perform *k-means* clustering with $k = 50$ to get the color centers. To compute the texture centers, the images in the dataset are first convolved with a MR8 (Leung & Malik, 2001) filter bank, consisting of 38 filters and 8 responses capturing edge, bar and rotationally symmetric features. After this step, the process is similar to computing the color histograms: we sample 50,000 points from the images, each point being 8-dimensional, and cluster them into 50 centers representing the texton codebook. We repeat this process of computing the color and texture clusters for each multi-domain setting separately. For each latent appearance vector $y_i$, we generate a source image $I_s = G(x_i, y_i, b_i, z_i)$ and a target image $I_t = G(x_j, y_i, b_i, z_i)$ such that $x_i, y_i$ belong to one domain and $x_j$ to another (e.g. $x_i/x_j$ represents bird/car shape). We then use a semantic segmentation network (DeepLabv3 (Chen et al., 2017)) pre-trained on MS COCO (Lin et al., 2014) to extract foreground pixels, and compute and compare color/texture based histograms between source and target images. Note that $I_t$ is likely to represent some visual concepts which were never seen by the segmentation network pre-trained on real data (e.g. car with dog's fur in Fig. 6), which could result in unreliable segmentations from the network. We circumvent that by generating an intermediate image $G(x_j, y_j, b_j, z_i)$ such that both the appearance/shape factors belong to the same target domain. We then run the pre-trained segmentation network on this image, and use the predicted foreground locations for $I_t$ (which has the same shape factor, $x_j$).

### A.4.2 SHAPE DISENTANGLEMENT

As mentioned in Sec. 4.2.1, one can compute the standard deviation image across the stack, apply an appropriate threshold (we use values between 0.1-0.3) to get a binary mask mostly focusing on the foreground, as shown in Fig. 11

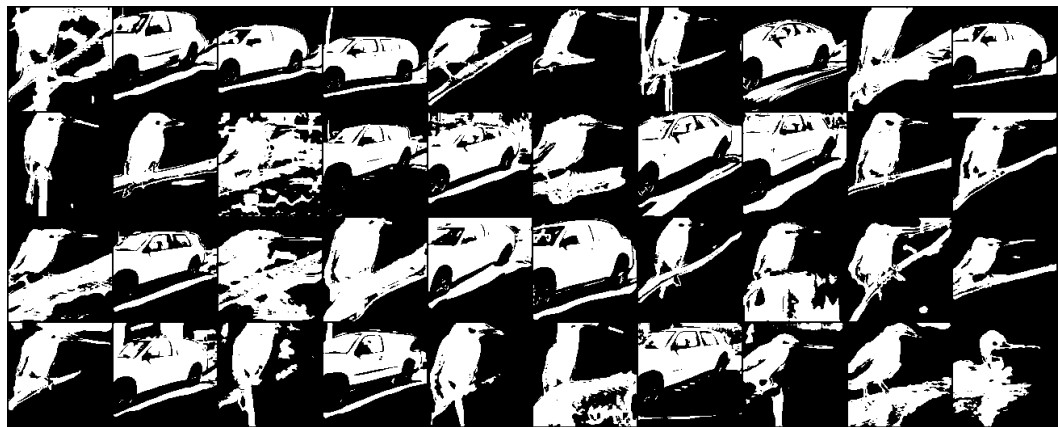

Figure 11: Sample binary masks generated on the standard deviation image for different shapes, by applying appropriate threshold

### A.4.3 EVALUATING PART CORRESPONDENCES IN APPEARANCE TRANSFER

In Sec. 2, we discussed that when the two domains under consideration have part correspondences (e.g. eyes/mouth of dogs and fox), transferring appearance from one to another happens in a way where properties around common object parts remain similar. We illustrate examples of this in Fig. 6 animals $\leftrightarrow$ animals, where (i) the brown patches near the eye of the dog get transferred to the similar region in husky (second last row), (ii) the brown patch near the cheeks of the dog get transferred to the cheeks of the leopard.

In this section, we study this property empirically. Given an input image, we first extract six keypoints (nose, two eyes, forehead center, two ears) using an animal face keypoint detector. For each such keypoint, we consider a small patch of 16 x 16 around it, and compute the color/texton histogram for this region. The process is similarly repeated for all the keypoints, resulting in six color histograms. When this appearance is transferred to a new image, the process is repeated again, by getting six histograms for the output. Finally, we compare the color histograms of corresponding keypoints in input/output using $\chi^2$-distance. If the part-level correspondences are preserved during the appearance transfer, then this distance should be low. The overall process of computing and comparing the histograms is similar to one described in Sec. A.4.1. However, since the animals $\leftrightarrow$ animals setting consists of a large diversity of species, there are cases where the pre-trained keypoint detector exhibits low confidence while making predictions. We choose to not consider these cases during the evaluation, since it could negatively impact a method's performance.

For color histogram similarity, FineGAN gets a score ($\chi^2$-distance) of $0.6197 \pm 0.26$, whereas our method gets $0.5404 \pm 0.27$. The trend holds for texton based histograms as well, where FineGAN gets a score of $0.7851 \pm 0.22$, and our method gets $0.7178 \pm 0.24$. So, in general, our method results in a transfer where color/texture frequency around points of interest (e.g. eyes) remain more similar than FineGAN.

### A.5 UNDERSTANDING HYBRID IMAGES THROUGH IMAGE CLASSIFIERS

A high level goal of this work was to create hybrid, out-of-distribution images which do not exist in any domain exclusively. We studied our method's ability to do so in terms of its ability to accurately transfer properties (shape/appearance) across different domains. Orthogonal to this, however, could we use image classification as a proxy to understand *hybridness*? Let us look at one of the results of FineGAN, from Fig. 6 row 4, dogs $\leftrightarrow$ cars. The resulting car generated upon transferring the appearance of dog onto a car still looks like a *realistic* car, i.e. could potentially belong to the real dataset of cars.

We posit that our method better transfers the semantics of the dog to a car; consequently, we expect the hybrid car and the dog to be more similar in some semantic feature space using our method, compared to FineGAN. We test this as follows - given an in-domain image ($x$ and $y$ are tied) $I =$

|  | $\mathrm{sim}(I, I_a)$ | | $\mathrm{sim}(I, I_s)$ | |
|---|---|---|---|---|
|  | **FineGAN** | **Ours** | **FineGAN** | **Ours** |
| birds $\leftrightarrow$ cars | $0.2877 \pm 0.035$ | $0.3182 \pm 0.040$ | $0.6787 \pm 0.095$ | $0.5959 \pm 0.084$ |
| cars $\leftrightarrow$ dogs | $0.2976 \pm 0.095$ | $0.3624 \pm 0.136$ | $0.6067 \pm 0.157$ | $0.4473 \pm 0.110$ |
| dogs $\leftrightarrow$ birds | $0.3669 \pm 0.081$ | $0.4209 \pm 0.077$ | $0.5315 \pm 0.089$ | $0.5081 \pm 0.095$ |
| animals $\leftrightarrow$ animals | $0.4618 \pm 0.090$ | $0.6375 \pm 0.136$ | $0.6850 \pm 0.0963$ | $0.4981 \pm 0.101$ |

Table 3: Assessing hybrid image properties through image classification. A high $\mathrm{dist}(I, I_a)$ score implies better transfer of semantic properties through appearance transfer. A low $\mathrm{dist}(I, I_s)$ score implies that appearance transfer on the same shape resulted in a shift away from the original distribution.

$G(x, y, b, z)$ (real brown dog), we combine its appearance ($y$) with factors from other domain to create a hybrid image, $I_a = G(x^{'}, y, b^{'}, z)$ (the hybrid car). We pass both these images through an image classifier (we use VGG 16) pretrained on ImageNet Deng et al. (2009), and compute cosine similarity between the feature representation of these two images. A high similarity implies better transfer of semantics, i.e. a better hybrid image. We repeat this experiment for all the in-domain images, and present the averaged results in Table 3 (left); we see that our method consistently achieves higher similarity, especially for cars $\leftrightarrow$ dogs and animals $\leftrightarrow$ animals.

Analogously, if transferring the dog's appearance did make the car more hybrid, its similarity with the original car should decrease. That is, given the same in-domain image $G(x, y, b, z)$ (original car), we borrow appearance from the other domain $y^{'}$ to create the hybrid image $I_s = G(x, y^{'}, b, z)$. We repeat the previous experiment and get the cosine distance between their feature representation. Lower similarity implies that the hybrid image *shifted away* from the original image. Averaged results (over all in-domain images) are presented in Table 3 (Right). We again observe that our method gets lower similarity score, implying more shift from a domain towards hybrid distribution.

## A.6 QUALITATIVE RESULTS

Fig. 12 demonstrates some more results from our method on different multi-domain settings discussed so far.

## A.7 COMPARISONS TO STARGANV2

Fig. 14 illustrates the results of StarGANv2 and our method. The key difference here is the properties that a method learns to disentangle: StarGANv2 can take the pose (the original paper uses the term "style" for this) of an animal (source image), and transfer it to a different animal (reference). So, in the resulting generated image, *both* animal's shape and appearance come from one image (reference) and only pose comes from a different one. For example, the lion (last row) when borrowing properties from different animals still remains a lion while changing poses: it's shape doesn't become that of a cat (first column).

Our method goes a step further, where it can also disentangle shape from appearance. For example, when transferring the appearance of a tiger to a dog (second column), the resulting image precisely preserves the dog shape (e.g. its signature long ears, tongue), and gets the tiger's appearance. So, our method has more control over different factors of an image.

We further study StarGANv2's disentanglement abilities, by using birds, cars and dogs as the three domains in the image-to-image translation setup. The question we ask is the following: can the model learn to combine a source bird and a reference car, so that the resulting generation has the shape of the bird but appearance of the car? Fig. 13 illustrates many such {source, reference} combinations which are used as input. We observe that the reference image dictates most of the properties of the generated images, with the style image only (slightly) altering the object pose. The results in this case again demonstrate that StarGANv2 has issues capturing shape from one domain (e.g. style image) and appearance from another (e.g. reference image). In contrast, we present our method specifically for this purpose, where the results shown in Fig. 12 and 6 demonstrates its ability in learning inter-domain disentanglement of shape and appearance.

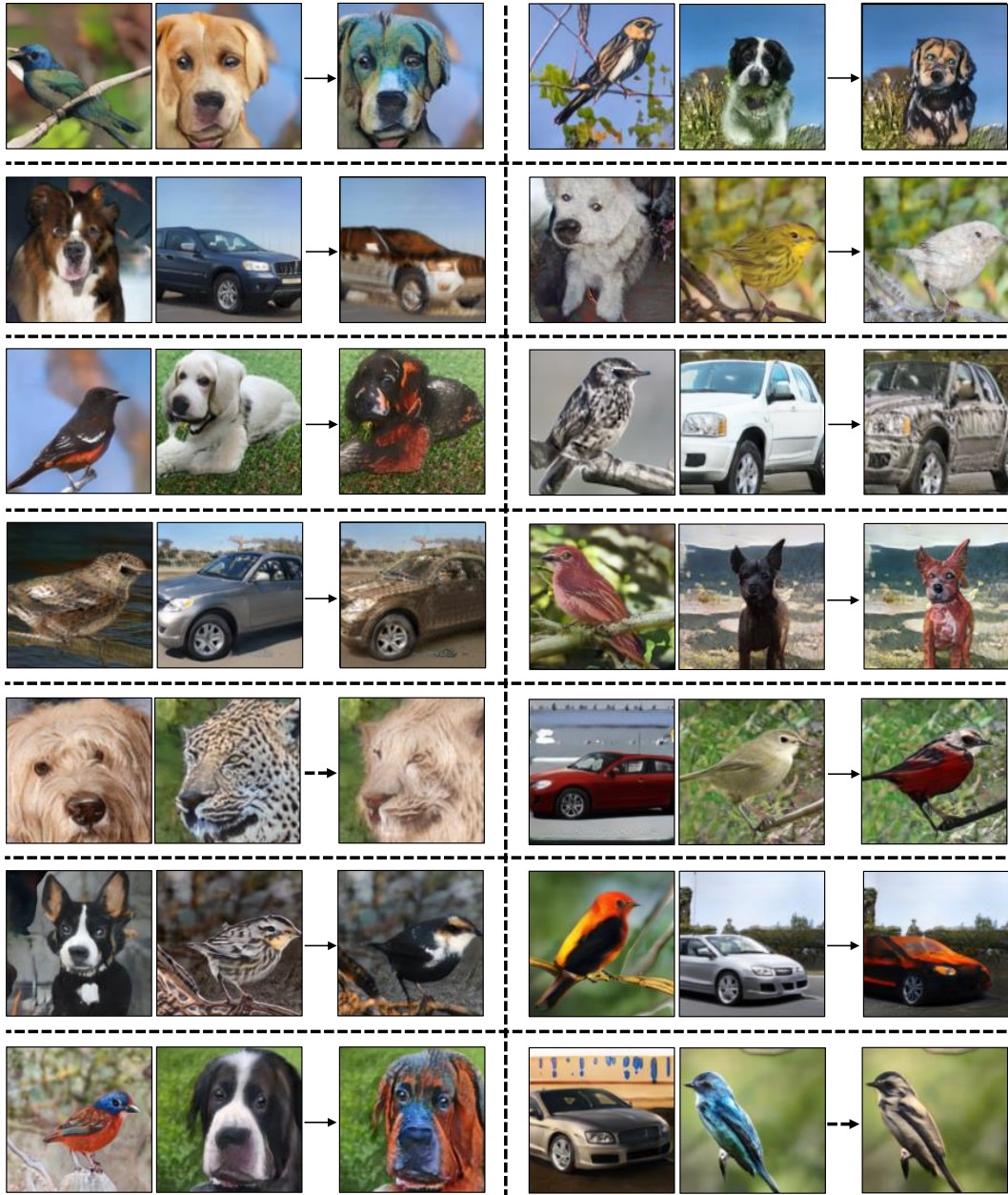

Figure 12: Each cell block follows `[Appearance, Shape → Output]`. Assorted results from different multi-domain settings using our method.

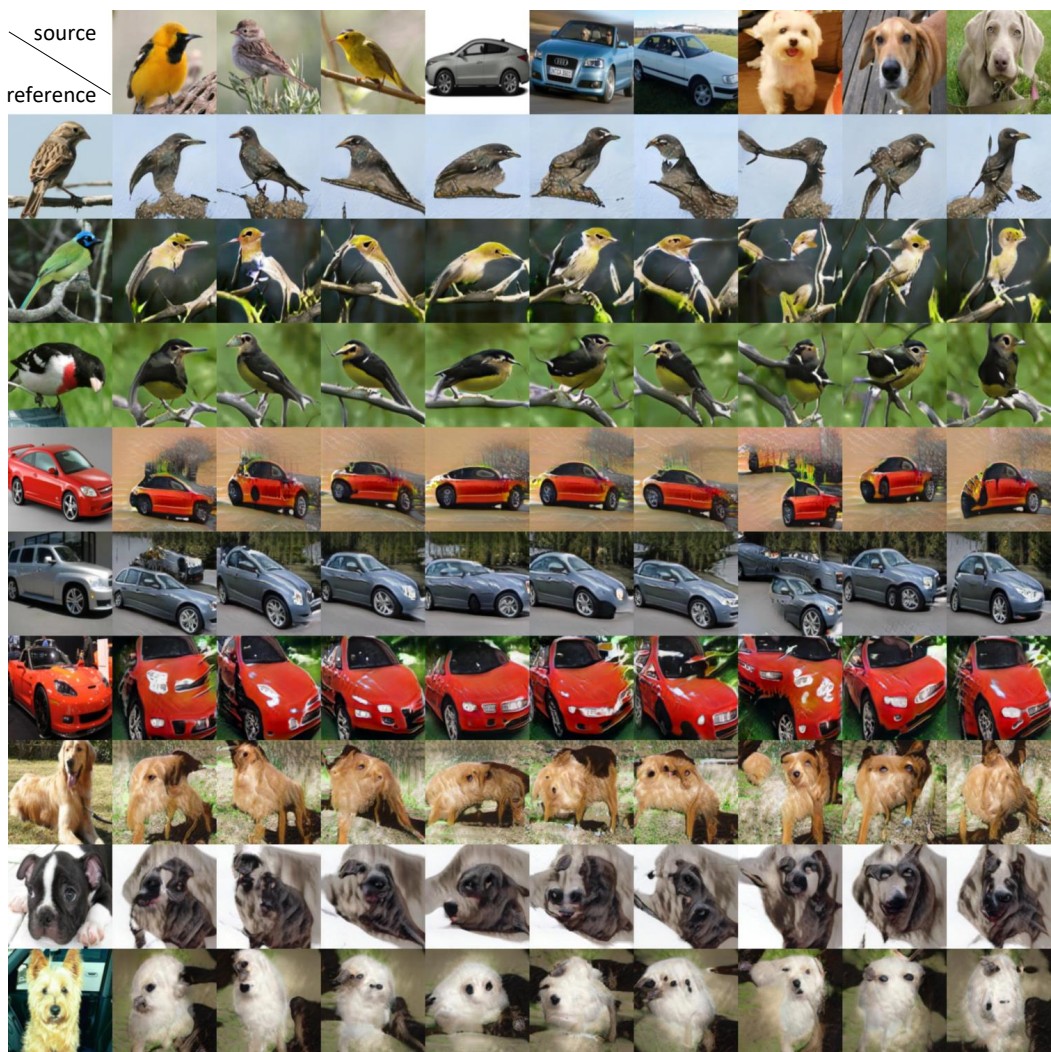

Figure 13: StarGANv2 results on using birds, cars and dogs as multiple domains. The images in the topmost rows and leftmost columns are real input images, while the rest are generated by the algorithm.

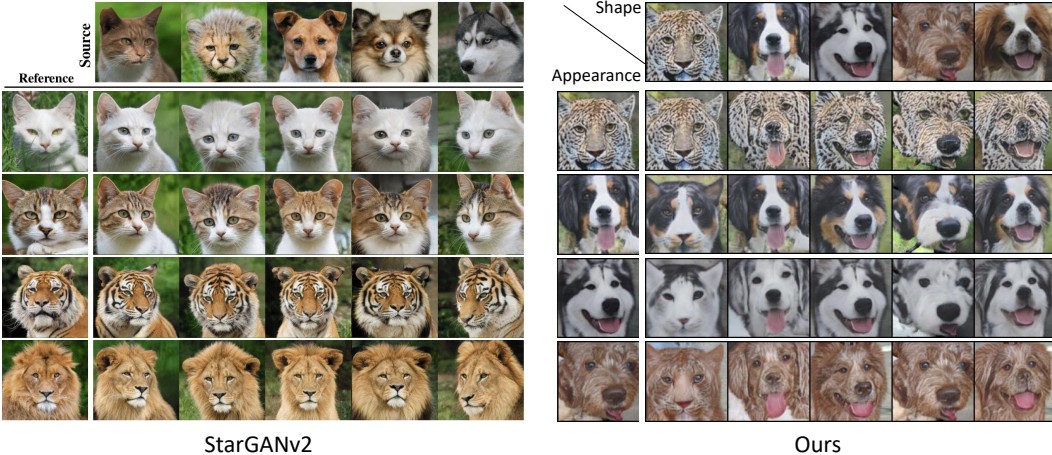

Figure 14: Comparing the disentanglement learnt by StarGANv2 (Choi et al., 2020) and our method.

