# OpenReview forum: "Generating Furry Cars: Disentangling Object Shape and Appearance across Multiple Domains"
_ICLR.cc/2021/Conference — ICLR 2021 Poster_

### Official Review · AnonReviewer2 · 2020-10-28
**Interesting direction, limited contribution**

**Rating:** 5
**Confidence:** 4

**Review:**

The submission describes a method to disentangle shape and appearance of images across two domains such that new images can be generated that have appearance and shape from either of these domains while still being visually convincing. Starting from an established method (FineGAN) to disentangle shape, appearance, background identity as well as a set of "nuisance" factors such as pose in one domain, the paper proposes to add a loss term that aims at retaining appearance when moving from one domain to another. This additional loss term essentials tries to keep the low-level image statistics between two images when both of them are generated with the same appearance, but possibly different shapes. It is trained such that it is invariant to the nuisance parameters (same object under differing views has same statistics), but discriminative towards the object appearance (different objects from the same view have differing statistics). The low level features are expressed as histograms of responses of convolution filters over the masked foreground pattern. The paper provides empirical evidences in the form of example images where appearance and shape are combined from two different domains (out of cars, birds, dogs, animals) as well as proxy measurements for the quality of the transfer: (a) how much do the low-level statistics differ in terms of $\chi^2$ distance, (b) how well is shape disentangled under changing appearance by measuring the foreground overlap between samples, (c) a user preference study (which method transfers shape and appearance better?). The results are compared to some relevant baselines (FineGAN, CycleGan, AdaIn, MUNIT), and show moderate improvements over those.

### Strengths
**[S1]** The paper is written well and it seems that one could reproduce the method reasonably.

**[S2]** The authors lay out their claims clearly.

**[S3]** Based on the given motivation, retaining the low level statistics of the image, the authors derive how to model a solution, optimize and evaluate it with respect to this motivation in a structured manner.

**[S4]** The empirical evaluation seems to demonstrate the effectiveness of the proposed solution towards the posed objective.

**[S5]** The authors allude to the method being able to translate appearance under shape change when there is either part-part correspondence between the domains (dogs$\leftrightarrow$cats) or none (dogs$\leftrightarrow$cars). However, see [W2,W3].

### Weaknesses
**[W1]** While the overall motivation is fairly clear: How can I retain more appearance between domains where I do not have access to actual labeled samples for training?, the particular heuristical choice of approach, retaining the frequency statistics of low-level filter bank responses, is not well motivated. Low level filter bank response statistics are known to encode texture-like properties in the sense of repeatable patterns such as a cheetah's fur texture (Figure 5, rightmost panel). Are they sufficient to capture other properties of appearance that are less readily encoded in low-level frequency statistics? What other choices of embodying appearance, including texture, are there, and why is the choice of the presented low-level statistic favorable? I feel that the paper is lacking in setting the heuristic in context so that the reader can be confident in the choice of heuristic.

**[W2]** The definition of intra- vs. inter-domain seems somewhat vague. It would be helpful to characterize this distinction either more theoretically or empirically: When are two domains close enough so that I do not need the additional term, when are they too far apart for even this method to work? Cats and dogs compared to cats and cars seem qualitatively different concept relations. The authors do imply hierarchies of closeness of domains (e.g. Section 2: having part level or no part level correspondences), but this is not used in the paper to clearly design or evaluate the method with respect to differing inter/intra-domain distributions or levels of domain proximity.

**[W3]** Weakly supported claim: Section 2 claims "Moreover, when part-level correspondences do exist (e.g., dogs ↔ tiger), it combines appearance and shape in a way which preserves them." This indeed could be a strong point, see W2. As I read the paper however, this claim seems only anecdotally supported, e.g. by individual samples in figures 5/6. I feel that more thorough investigation and evidencing of this claim would strengthen the paper.

**[W4]** Reference missing: I feel that [B1] below does indeed discuss and investigate related concepts, albeit from the perspective of style rather than object appearance. Nevertheless, it discusses disentanglement under disjoint domains, and it would be interesting to include this in the discussion of related work.

**[W6]** I read the submission as more of an empirical paper than theoretical paper. From this perspective, the presented empirical evidence seems limited. In order to gauge the effectiveness and benefit of the method better, I wished for a wider range of data, where I can see the relation of shape/appearance transfer and the "distance" of the domains better. The authors do mention, but do not show for instance furniture. Can I transfer from cups to people or vice versa?

### Further comments
**[C1]** There is a flipped $\pm$ in table 2 ("Ours vs. FineGAN / dogs $\leftrightarrow$ birds")

### Summary
I feel that there is benefit to the direction that the submission is taking. Although at this point the weaknesses outweigh the strengths, a revision could make a strong contribution if for instance the choice of heuristic is motivated and evidenced more strongly in the context of potential alternatives, and the intra-/inter-domain distinction is worked out more clearly theoretically and/or empirically. As it is I feel that the paper would need significant revision for acceptance at ICLR.

**[B1]** @InProceedings{Lee_2018_ECCV,
author = {Lee, Hsin-Ying and Tseng, Hung-Yu and Huang, Jia-Bin and Singh, Maneesh and Yang, Ming-Hsuan},
title = {Diverse Image-to-Image Translation via Disentangled Representations},
booktitle = {Proceedings of the European Conference on Computer Vision (ECCV)},
month = {September},
year = {2018}
}

---

> ### Author Response · Authors · 2020-11-23
> **Authors' response**
>
> Thank you for your time and feedback. We address the mentioned concerns below. All references to figures and sections are based on our revised paper pdf.
>
> **Choice of heuristic:**
>
> The key property that we desire out of our heuristic is that it should not capture any shape information, and only encode appearance properties. Capturing the appearance through a low-level, frequency-based representation helps satisfy this aspect. It ensures that, for example, when transferring the appearance of a car to a dog, we only transfer color/texture blobs, and don’t transfer higher level concepts (e.g., wheels). Furthermore, our method can capture the low level features across a range of spatial resolutions. Figure 4 depicts a special case of our method, where only one layer of convolutional filters (i.e. filter bank) is used to approximate the frequency based appearance representation; however, we can easily stack additional convolutional layers, and use their responses as well while computing the appearance representation (Appendix A.2 - “Receptive field size of the filters”). This increases the effective receptive field of filters in deeper layers, which results in capturing bigger color/texture blobs. The following examples illustrate this point: (i) in Figure 1 (bottom left), the black beak and neck region of the bird; (ii) in Figure 2, the blue/red color blocks in the bird; (iii) in Figure 6 (‘ours’; dogs to cars; 2nd row), the yellow/black blocks in the car; (iv) in Figure 6 (‘ours’; animals to animals; last row), the brown/black patches of the dog.
>
> **Intra vs inter domain clarification:**
>
> We define domains to correspond to “basic-level categories” [1]; e.g. birds vs dogs vs cars. With this definition, images belonging to the same domain typically have more visual similarity than those belonging to different domains. As such, intra-domain disentanglement is an easier task compared to inter-domain disentanglement, since it concerns a single domain, where images are more similar to each other. We agree that certain pairs of domains could be closer than others, e.g. dogs are more similar to cats than cars, which can even have part-level correspondences. We empirically study these settings, as suggested by the reviewer (please see the next section).  We also admit that the boundaries between basic-level categories can be fuzzy (e.g., is a rickshaw a car or a motorbike or neither?) However, the main advantage of our approach is that it is domain agnostic: one does not need to worry whether the two domains are similar or not, as our method can work for both the intra-domain and inter-domain settings.
>
> Regarding the scenarios where our method wouldn’t work, as explained in Section 3.2, our base model FineGAN makes certain assumptions about the data (existence of a hierarchy between shape and appearance factors) to learn intra-domain disentanglement of factors. Since our method builds upon FineGAN, it is suited for datasets which possess those properties (e.g. birds/dogs), and won’t be as effective otherwise.
>
> **Part-level correspondences preserving appearance:**
>
> We agree that this claim would benefit from more evidence. To this end, we have included a section in the appendix (A.4.3), where we quantitatively study the effect of transferring appearance when two domains have part-level correspondences, i.e. the animals dataset. In summary, we detect 6 keypoints (nose, two eyes, forehead center, two ears) in the source image, and compute the color histogram for a patch around each of them. When transferring the appearance to another animal, we detect the new keypoints, and see if it has a similar color/texture histogram around the corresponding keypoints. For example, in the second last result of Figure 6 (Ours; animals to animals), the dog from which appearance is borrowed has brown patches over the eyes. Hence, when transferring it to a husky’s shape, the resulting animal should also have brown patches over the eyes, i.e. a similar color/texture distribution around the eye keypoint. We observe that our method better preserves this property compared to the baseline, FineGAN.
>
> **DRIT reference missing:**
>
> Thank you for bringing this work to our attention. We have updated the related work section to include its discussion and high level comparison with our problem setting. In summary, the attribute conditioned image-to-image translation application explored in DRIT was limited to cases where the attribute image shares some structure with the content image (e.g., natural and sketch images of face domain as content/attribute images respectively). In this work, we aim to combine factors from entirely different domains, having no structural similarity.
>
> **References**
>
> [1] Principles of categorization. Cognition and Categorization. Eleanor Rosch, pages 27–48, 1978

---

> > ### Author Response · Authors · 2020-11-23
> > **Authors' response (continued)**
> >
> > **Empirical evidence seems limited:**
> >
> > First, we respectfully disagree that the empirical evidence is somewhat limited, since we present experiments on various challenging multi-domain settings, and demonstrate the effectiveness through several measures, including user studies, as shown in Tables 1, 2, 3 and Figures 1, 5, 6, 7, 8, 9, 10, 11, 12, 13, and 14.  Second, the reviewer implies that the datasets used in the paper don’t have enough variety, and consequently one cannot properly study the distance between two domains. We would like to clarify that our goal with this work is not to study how close two domains are: it is to learn a disentangled representation of shape and appearance “regardless” of how similar (or dissimilar) two domains are. As for showing the results on more arbitrary datasets (e.g. humans/cups), we believe that combining factors from a bird and a car (or a car and a dog) is similarly difficult as combining factors from a human and a cup, and hence believe that the dataset settings used for this work do offer significant challenges for a method to succeed.
> >
> > **Improvements are moderate over the baselines:**
> >
> > We respectfully disagree. First, baselines like CycleGAN/MUNIT/AdaIn illustrate clear issues while solving the problem at hand (Figure 5). We have also included a new comparison to StarGANv2 (Figures 13, 14) which again show clear limitations. Second, compared to FineGAN, we achieve superior results qualitatively (Figure 6) and quantitatively (Tables 1 and 2). It would be helpful if the reviewer could point out specific instances where our improvements over FineGAN (or any other baseline) appear moderate.

---

### Official Review · AnonReviewer3 · 2020-10-28
**Slightly novel approach, but need to improve experiment results**

**Rating:** 5
**Confidence:** 4

**Review:**

--Summary:
The paper proposed a method to learn disentangled representation of shape and appearance for cross-domain (different object categories) data. Build upon FineGAN, the method uses contrastive learning combined with normalized temperature-scaled cross-entropy loss to further disentangle the shape and appearance information.

--Strongness:
1. The model is slightly novel. They combine contrastive learning with normalized temperature-scaled cross-entropy loss to learn the filter bank to construct the appearance feature histogram.
2. They perform many experiments including comparisons with baselines and ablation studies on the proposed loss terms. They demonstrate the effectiveness of their approach to generating hybrid images.
3. The paper is well-organized.

--Weakness:
1. The motivation is still unclear. I still don't get the point for the usefulness of appearance transfer across two different types of objects (e.g. car and animal) which they claim as their contribution. For example, I don't see the application for applying car appearance to animals.
2. The comparison baselines are too old. For the appearance transfer comparisons as shown in Figure 4 are the papers before 2018. For example, why don't you compare your model with StarGANv2[1] which also demonstrates appearance transfer to different shapes (e.g. Figure 10)?

--Questions:
1. I'm curious about the results if you replace the histogram method by just using CNN to extract features on the masked output from FineGAN?

--Recommendation:
Although the authors demonstrate the effectiveness of the proposed method, there are some concerns to be addressed:
1) Motivation is not intuitive.
2) There are many more recent papers for transferring appearance to another shape, e.g. StarGANv2[1], which is not included in the experiment.

I currently vote negatively but the authors are strongly encouraged to address these concerns.

[1] StarGAN v2: Diverse Image Synthesis for Multiple Domains, CVPR'20

---

> ### Author Response · Authors · 2020-11-22
> **Authors' response**
>
> Thank you for your time and feedback. We address the mentioned concerns below. All references to figures and sections are based on our revised paper pdf.
>
> **Comparisons to recent image-to-image translation methods (e.g., StarGANv2):**
>
> We agree that such comparisons would strengthen the results. We’ve hence included StarGANv2 (as suggested by the reviewer) in the related work section, and an empirical comparison in the appendix (A.7) for different datasets. In sum, StarGANv2 struggles to disentangle shape and appearance when the domains are very different (see Figure 13). Even for domains that are similar (animals $\leftrightarrow$ animals; see Figure 14 left), StarGANv2 is only able to disentangle pose from the other factors (shape and appearance); i.e., when combining the properties from a source and reference image, the object shape and appearance factors are taken from the reference image, and only the object pose factor is taken from the source image (so that a tiger’s pose changes to match the pose of a dog). In contrast, our approach can disentangle shape from appearance so that e.g., a dog is texturized with a leopard’s texture (see Figure 14 right).
>
> **Motivation not clear:**
>
> We discuss some possible applications of our work in the last paragraph of Section 1. We provide more examples here. Let’s consider the virtual try-on application, where people virtually try different kinds of clothes over the same body figure. One of the desired features is, for example, to keep the same t-shirt type (e.g. round-necked), but borrow it’s appearance from a different t-shirt (with collar). So, the options available to choose the appearance from is limited to the domain of t-shirts. But what if one sees a leopard and wishes to try on a t-shirt with its appearance?  What if one wishes to imagine how a handbag would look with a golden retriever’s fur texture? The reviewer points out that there isn’t much application of applying a car’s appearance to a dog. We agree, but we believe there could be an application for the inverse task: imagining a car (or other man-made objects, e.g. shirts, handbag, furniture) in the appearance of other arbitrary objects (e.g. dog, leopard). The fashion/design industry has been evolving, where people are increasingly interested in unorthodox design choices for a variety of man-made objects. With this work, we aim to provide users with the "option" to borrow appearance from a much wider variety of objects (Section 1, last paragraph).
>
> Furthermore, we believe that learning disentangled representations for shape and appearance across multiple domains serves as a more challenging and accurate testbed for evaluating disentanglement. As mentioned in Section 1, last paragraph, if shape and appearance can be combined from different objects within a domain, but not when they belong to different domains, then those factors are not completely independent of each other, and hence not accurately disentangled.
>
> **Using CNN to extract features:**
>
> The filter banks used to extract appearance features, as illustrated in Figure 4, are in fact implemented using convolutional layers.  Section A.2 in Appendix, “Receptive field size of filters”, provides further details and also experiments with different variants, by using 1 or 2 or 3 layer convolutional networks to extract the appearance based features for constructing the histograms. Furthermore, by computing a histogram-based representation, we remove the spatial structure of the CNN features for an input image, and as a result, the comparison between images is done based only on low-level appearance feature statistics (and not on shape-level information). If we were to instead directly compare the CNN features, then $L_{hybrid}$ would constrain not only the images' appearance feature statistics to be similar, but also their shape features to be similar.

---

### Official Review · AnonReviewer1 · 2020-10-29
**Good results, but paper could benefit from clearer explanation**

**Rating:** 7
**Confidence:** 4

**Review:**

The authors build upon prior work (FineGAN) in intra-domain disentanglement to extend to inter-domain transfer of separate attributes. Since no ground truth data exists for inter-domain transfer, they use contrastive losses to enforce similar statistics of low-level filter activations (averaged over the image) as a proxy for appearance similarity.

Appearance transfer experiments on performed on a good selection of datasets and the results of the proposed method represent a qualitative step forward in unsupervised conditional generation across domains. Quantitative metrics support this point.

That said, the paper could be significantly improved by spending less space motivating and defining the problem, and more space describing the actual method used. The authors mention FineGAN in passing as their base model, but it is essential to the proposed method and could use further elaboration. As is, the relevant details of the losses and architecture choices are not contained within the paper itself. For instance, the loss terms in L_{base} are not defined and no diagrams are given to help understand the workings of the base model.  Similarly, the training procedure is a bit unclear from the text. If the content of Figure 3 were expanded, or a training algorithm table provided, even in the supplemental it would significantly improve the paper by not relying on a reference to provide the description of the core technique.

---

> ### Author Response · Authors · 2020-11-22
> **Authors' response**
>
> Thank you for your time and feedback. All references to figures and sections are based on our revised paper pdf.
>
> **More space describing FineGAN:**
>
> We agree that the paper could benefit from more details about FineGAN, our base model. We have hence updated Section 3.2 to include more details about FineGAN’s loss functions that constitute $L_{base}$ (their equations and precise inputs), its overall image generation process, and a new figure (Figure 3 left) illustrating its model architecture. We would be happy to further revise the content if needed.

---

### Official Review · AnonReviewer4 · 2020-10-30
**Nice extension of shape/appearance disentanglement to images from different domains**

**Rating:** 7
**Confidence:** 3

**Review:**

### Summary
This paper proposes a generative model as an extension of FineGAN that aims to learn a disentangled representation for image shape and appearance across different domains rather than "intra-domain" disentanglement. To this end, the authors adopt the prior that features that correspond to an object's appearance should preserve frequency histograms. In order to incorporate this prior into the differential learning procedure, they learn a library of convolutional filters using a contrastive learning framework. They provide many convincing baselines and comparisons to related work and are able to attain reasonable results for style/content transfer between unrelated domains.

### Explanation of rating
I think this paper is a good steps towards truly being able to learn generic disentangled representations. Although the kind of data used for experiments is relatively simple, the results that are achieved go beyond existing state-of-the art generative models.

### Pros
- This provides some new insight into the kinds of disentanglements that previous generative models are (and are not) able to discover.
- The frequency histogram assumption is a nice prior that is general enough to apply to different domains
- The evaluation and comparisons are quite extensive and convincing.

### Cons/questions
- It might help to clarify and emphasize the novelty of the proposed method vs. the parts of FineGAN that it builds upon. For instance, the authors claim that that their method supports intra-domain disentanglement. While this is true, it seems like this is a feature of the base model and not really a contribution.
- All of the results shown involve images with a single subject that takes up most of the canvas? How does this behave on less obvious images, e.g. with less prominent or multiple subjects?
- How are $N_x$, $N_y$. $N_b$ chosen?
- How is sim in eq. 1 defined?
- What is temperature $\tau$ in eq. 1? How is it chosen / is the method stable to choice of temperature?
- Page 1: "it's appearance" -> "its appearance"
- Page 2: "acros" -> "across"

---

Thanks to the authors for the clarifications. I have read the other reviews and responses and still believe that the paper is a good contribution. Therefore, I am keeping my score.

---

> ### Author Response · Authors · 2020-11-22
> **Authors' response**
>
> Thank you for your time and feedback. We address the mentioned concerns below. All references to figures and sections are based on our revised paper pdf.
>
> **Novelty of the proposed method vs. the parts of FineGAN it builds upon:**
>
> Our key novelty over FineGAN is that we learn disentangled representations across multiple domains, so that we can combine one factor from one domain (e.g. shape of a car) and another factor from another domain (e.g., appearance of a bird), as mentioned in Section 1 and Figure 2. FineGAN, on the other hand, only addressed learning disentangled representations within a single domain (Figure 3).
>
> In terms of the approach, we build upon the architecture of FineGAN (Figure 3), and introduce additional layers of filters that are designed to capture a low-level frequency based representation of an object (Figure 4a). To enable this, we introduce additional objective functions ($L_{filter}$ and $L_{hybrid}$), which use contrastive learning to learn the desired filters, and consequently enforce the model to better disentangle shape from appearance.
>
> **Behavior with less prominent or multiple subjects:**
>
> The base model, FineGAN, is designed to disentangle factors for images that contain a single foreground object. Since we build upon FineGAN, we are also limited in the same way.  Thus, although the multiple subjects scenario is interesting, it is beyond the scope of this work.  In order to extend the approach to work for multi-subject images, we would need to have at a minimum separate latent codes for each object (e.g., a separate background/shape/appearance/pose code for each object in the image).
>
> As for less prominent objects (i.e., object area small compared to overall image area), our paper does present results where appearance is being borrowed from a relatively small-sized object. For example, (i) Figure 6 (Ours, birds to cars, third row): the yellow bird only occupies a small area of the overall image; (ii) Figure 12 (first row, first column): the greenish/bluish bird forms a small part of the image, but our method still transfers its appearance accurately to the dog.
>
> **How are $N_x$, $N_y$, $N_b$ chosen:**
>
> Section A.1 (paragraph 1) discusses the values used for $N_x$, $N_y$, $N_b$ for different domains. We’ve further added more details pertaining to the choice of these hyperparameters. In sum, we set $N_x$ and $N_b$ to be the total number of ground-truth fine-grained (subordinate) object categories across the two domains, and $N_y$ to be 1/10 of that number.  This follows what was done in the FineGAN paper, which also found that the disentanglement results are largely agnostic to these specific choices.
>
> **How is ‘sim’ defined in eq. 1?**
>
> sim refers to cosine similarity. We have updated the text to include this detail.
>
> **Information about the temperature parameter:**
>
> We updated the paper to provide the temperature value, below Equation 1. In our initial experiments, we used the same value of 0.5 for the temperature hyperparameter for all experiments. In Section A.2 (Effect of temperature), we conducted additional experiments where we vary the temperature value. In sum, our method is largely independent of the specific choice of temperature value.

---

### Author Response · Authors · 2020-11-24
**Updates to the submission**

We thank the reviewers for their thoughtful comments. We have provided direct and detailed responses to each reviewer below.  We have also updated our submission (changes denoted in red), which we summarize here:

- More details about the base model, FineGAN, that we build upon, including its architecture, loss functions, and the overall image generation process (Section 3.2 and Figure 3).
- Comparison to StarGANv2 on various datasets (Appendix A.7 and Figures 13, 14).
- A discussion on DRIT (Section 2).
- Empirical analysis on how our model’s appearance transfer respects part-level correspondences between two related domains (Appendix A.4.3).
- More details and ablation studies on some of the hyperparameters we use in our experiments: how we choose $N_x$ (number of shapes) and the temperature parameter (Appendix A.1, A.2 and Figure 9).

---

### Decision · Program_Chairs · 2021-01-07
**Final Decision**

**Decision:**

Accept (Poster)

**Comment:**

The paper swaps characteristics of an object in one image onto those of another object in another image--for example, adding fur to a car.  The authors give some examples where the task could be useful.  Further, they successfully argue  that this task is an illustration that the disentanglement task has been done well.

Two reviewers argued for acceptance, two for just-below-the-bar rejection.  The 2nd of those in favor of rejection engaged thoughtfully with the authors and raised the score by 1 after that engagement.

We have decided to accept the submission as a poster.